# MicroRNAs down-regulate homologous recombination in the G1 phase of cycling cells to maintain genomic stability

Young Eun Choi[1], Yunfeng Pan[1], Eunmi Park[1†], Panagiotis Konstantinopoulos[2], Subhajyoti De[3], Alan D'Andrea[1], Dipanjan Chowdhury[1]*

[1]Department of Radiation Oncology, Division of Genomic Stability and DNA Repair, Dana-Farber Cancer Institute, Harvard Medical School, Boston, United States; [2]Department of Medical Oncology, Dana-Farber Cancer Institute, Harvard Medical School, Boston, United States; [3]Department of Medicine, University of Colorado School of Medicine, Aurora, United States

**Abstract** Homologous recombination (HR)-mediated repair of DNA double-strand break (DSB)s is restricted to the post-replicative phases of the cell cycle. Initiation of HR in the G1 phase blocks non-homologous end joining (NHEJ) impairing DSB repair. Completion of HR in G1 cells can lead to the loss-of-heterozygosity (LOH), which is potentially carcinogenic. We conducted a gain-of-function screen to identify miRNAs that regulate HR-mediated DSB repair, and of these miRNAs, miR-1255b, miR-148b*, and miR-193b* specifically suppress the HR-pathway in the G1 phase. These miRNAs target the transcripts of HR factors, BRCA1, BRCA2, and RAD51, and inhibiting miR-1255b, miR-148b*, and miR-193b* increases expression of BRCA1/BRCA2/RAD51 specifically in the G1-phase leading to impaired DSB repair. Depletion of CtIP, a BRCA1-associated DNA end resection protein, rescues this phenotype. Furthermore, deletion of miR-1255b, miR-148b*, and miR-193b* in independent cohorts of ovarian tumors correlates with significant increase in LOH events/chromosomal aberrations and BRCA1 expression.

*For correspondence: dipanjan_ chowdhury@dfci.harvard.edu

Present address: †School of Life Science and Nano-technology, Hannam University, Daejeon, Korea

Competing interests: The authors declare that no competing interests exist.

## Introduction

Double-stranded DNA break (DSB)s are deleterious for cell health and accurate repair of DSBs is critical for maintaining genome stability. Two major mechanistically distinct pathways, homologous recombination (HR) and non-homologous end joining (NHEJ) have evolved to repair DSBs (*Ciccia and Elledge, 2010*; *Chapman et al., 2012b*). HR requires an undamaged homologous DNA template to replace an adjacent damaged one with high fidelity (*San Filippo et al., 2008*). By contrast, the untemplated NHEJ pathway is more error-prone as it rapidly processes and joins broken DNA ends (*Lieber, 2010*). There is tight regulation of the DSB repair pathways during the cell cycle as HR is restricted to the S/G2 phase and NHEJ is pre-dominant in G1 but has moderate activity throughout the cell cycle. The balance of HR and NHEJ proteins (such as BRCA1 and 53BP1) involved in early steps of the two repair pathways are critical for pathway choice and the cell-cycle phase specific regulation of each pathway (*Bouwman et al., 2010*; *Bunting et al., 2010*; *Chapman et al., 2012b*). Initiation of HR via resection of broken DNA ends blocks NHEJ and could lead to unrepaired DSBs in G1 cells (*Helmink et al., 2011*; *Escribano-Diaz et al., 2013*). The physiological consequence of HR-mediated repair of a DSB in the G1-phase is the loss of heterozygosity (LOH).

The molecular mechanism of restricting HR to S/G2 phases of the cell cycle is of paramount importance in cancer biology. Ectopic HR leading to LOH may play a critical role in carcinogenesis as recessive oncogenic mutations are revealed and/or tumor suppressor function is lost (*Bishop and Schiestl, 2003*).

**eLife digest** The DNA in a cell is damaged thousands of times every day. One of the most serious types of damage involves something breaking both of the strands in the double helix. Such a double-strand break can delete genes or even kill the cell. In fact, conventional cancer therapy kills cancer cells by causing irreparable double-strand breaks. Conversely, a normal cell that is constantly exposed to DNA damaging agents can become a tumor if double-strand breaks are incorrectly repaired. An efficient and accurate double-strand break repair system needs to be in place to prevent this transformation. Therefore, an in-depth understanding of double-strand break repair and the factors involved are important for both gaining insight into the cause of cancer and to improve cancer therapy.

Cells have evolved several different ways to detect and repair double-strand breaks. A method called homologous recombination, for example, uses an undamaged DNA molecule as a template that can be copied to make new DNA. Since it needs a readily available DNA template, this method only works in phases of the cell growth cycle where there are many copies of DNA—that is, in the post-DNA replication phases. In particular, homologous recombination does not work during the pre-replication, G1 phase. If homologous recombination is attempted during G1, it will block the other methods employed by cells to repair broken strands of DNA.

An important challenge is to understand how homologous recombination is restricted to particular parts of the cell cycle. Although certain proteins associated with the early stages of double-strand repair are thought to determine the type of DNA repair that occurs, the details of this process are not fully understood.

One group of molecules that are thought to be involved are microRNAs, which normally limit the number of proteins produced from certain genes. However, since a single microRNA molecule can be associated with several proteins, and since a single protein can be associated with several microRNA molecules, it has proved difficult to establish the exact effects of a specific microRNA molecule.

Choi et al. now show that seven microRNA molecules can control homologous recombination, and three microRNAs in particular restrict homologous recombination during the G1 phase of the cell cycle. If these microRNAs are inhibited during the G1 phase, which allows homologous recombination to start, and counter-intuitively more double-stranded breaks are seen. However, if a gene involved in starting homologous repair—called CtIP—is silenced while the microRNAs are inhibited, then the DNA breaks are repaired. Exactly, how the microRNA molecules produce different effects during different phases of the cell cycle will be need to be investigated by future studies.

The regulation of the HR pathway is also extremely relevant for cancer therapy. Chemical inhibitors of the DNA repair protein, poly (ADP-ribose) polymerase (PARP), exhibit synthetic lethality in BRCA-deficient tumors that have a defective HR pathway (*Bryant et al., 2005*; *Farmer et al., 2005*). Although the molecular mechanism underlying this phenotype remains unresolved, lack of PARP activity in an HR-deficient scenario leads to the accumulation of DSBs in proliferating cells and this triggers apoptosis (*Helleday, 2011*). Epigenetic suppression of factors in the HR pathway occurs in sporadic breast and ovarian tumors, and collectively this phenotype has been described as 'BRCAness' (*Turner et al., 2004*). Tumors with the 'BRCAness' phenotype are likely to respond to PARP inhibitors, and identifying factors (such as microRNAs) that induce the 'BRCAness' phenotype may enhance the clinical utility and scope of PARP inhibitors.

MicroRNAs (miRNAs) are abundant small (~20–22 nts) non-coding RNAs that typically dampen gene expression at the post-transcriptional level (*Fabian et al., 2010*) and are aberrantly expressed in a variety of cancer cells (*Garzon et al., 2009*). The regulation elicited by miRNAs is highly complex, given that a single miRNA can influence the expression of many mRNAs, and conversely a single mRNA is targeted by multiple miRNAs (*Bartel, 2009*). Due to this complexity the role of miRNAs in DSB repair remains largely unknown (*Chowdhury et al., 2013*). Using the experimental system of in vitro hematopoietic cell differentiation, we discovered that miR-182 targets BRCA1 (*Moskwa et al., 2011*). Over-expression of miR-182 in triple negative breast tumor lines (in vitro and in vivo) suppresses HR via down-regulation of BRCA1 and sensitizes cells to PARP inhibitors. This served as a 'proof-of principle' that miRNAs may influence HR-mediated repair of DSBs.

In this study, we used PARP inhibitor sensitivity as a marker for HR deficiency to conduct a functional screen to identify miRNAs that down-regulate HR. Over-expression of seven miRNAs significantly reduced HR-mediated DSB repair. Based on their expression in a panel of breast and ovarian lines, we focused on characterizing the mechanism and physiological relevance of miR-1255b, miR-193b*, and miR-148b*. Despite lacking canonical binding sites, miR-1255b, miR-193b*, and miR-148b* associate with the BRCA1, BRCA2, and RAD51 transcripts and regulate their expression. Remarkably, the miRNA-mediated regulation of these genes is cell cycle dependent and inhibition of miR-1255b, miR-193b*, and miR-148b* leads to enhanced expression of BRCA1, BRCA2, and RAD51 specifically in the G1 phase. A functional consequence of inhibiting these miRNAs is a basal increase in DSBs in G1 cells. This phenotype can be reversed by silencing CtIP (the BRCA1-associated DNA end resection factor) which initiates HR-mediated DSB repair. Furthermore, deletion of these miRNAs in two independent cohorts of high-grade serous ovarian tumors correlates with increase in LOH.

## Results

### Screening miRNA mimic libraries for PARP inhibitor sensitivity

To systematically identify miRNAs that impact PARP inhibitor sensitivity, we screened two commercially available miRNA mimic libraries (Qiagen, Valencia, CA and Applied Biosystems, Grand Island, NY). miRNA mimics are 20–22 nt, chemically modified double-stranded RNA molecules designed to mimic endogenous mature miRNAs. The miRNA mimics from Qiagen and Applied Biosystems have proprietary chemical modifications that cause inactivation of the 'passenger' strand, allowing the 'active' strand to associate with target transcripts and regulate gene expression. The goal was to identify miRNA mimics that sensitize breast tumor cells, MDA-MB231 to the clinical PARP inhibitor, ABT888 (*Figure 1A,B*). The miRNA mimics from Applied Biosystems impacted ABT888 sensitivity over a broad range, ~10- to 12-fold difference between the mimics with the highest impact on viability and the control mimics. Whereas the viability range with the mimics from Qiagen was limited to 3- to 4-fold. BRCA2 siRNA served as a positive control for each plate and the mimic for miR-182 served as an internal control. BRCA2 is an important gene for repair of replication-induced DNA damage and depletion of BRCA2 has an impact on viability even in the absence of PARP inhibitors (*Figure 1—figure supplement 1*). miRNA mimics that caused significant loss of viability in the absence of PARP inhibitors (equal or more than BRCA2 siRNA) were not considered (*Figure 1—figure supplement 1*). The Applied Biosystems library yielded a rank ordered list of the top 13 miRNAs (including miR-182) based on viability percentage that sensitized cells to ABT888 (*Figure 1C*) ($Z$-score thresholds [$Z < -2$]). Prior to individual validation experiments, we utilized the results from the screen performed with the Qiagen library to confirm the impact of these miRNAs on ABT888 sensitivity. Our results show that 12 of the 13 miRNAs had at least a 25% impact on ABT888 sensitivity. Next, we confirmed the impact of these miRNAs on PARP inhibitor sensitivity using two clinical PARP inhibitors, ABT888 and olaparib (*Figure 1D*, *Figure 1E*) in two breast cancer cell lines, MDA-MB231 and 21NT (*Figure 1F*). Together, these results validate our primary screen and suggest that these miRNAs may regulate the efficacy of HR-mediated repair of DSBs.

### Impact of miRNAs on HR

Sensitivity to PARP inhibitors directly correlates with HR deficiency, but it is plausible that other pathways may also contribute to this phenotype. To test whether expression of the 12 selected miRNAs impacts HR, we used a functional assay that assesses HR-mediated repair of an I-SceI-induced DSB (*Nakanishi et al., 2005*). U2OS cells that have a single, stably integrated copy of the artificial recombination substrate DR-GFP were used for this assay. A chromosomal DSB was introduced in one of two tandem non-functional GFP genes by transient transfection of an I-SceI-encoding plasmid. A functional GFP gene can be reconstituted if the DSB is repaired by HR using the other partial GFP gene as a template. Using this assay, 11 of the 12 miRNAs significantly diminished HR-mediated DSB repair, which correlated with their impact on PARPi sensitivity. (*Figure 2A,B*). We prioritized the miRNAs according to their impact on HR and selected 7 miRNAs that reduced HR-efficiency by ≥30% (miR-1231, miR-1255b, miR-148b*, miR-876-3p, miR-221*, miR-193b*, and miR-185*). RAD51 is involved in HR, and quantification of nuclear RAD51 foci has also been used to evaluate HR in various cell systems including primary breast tumors (*Willers et al., 2009*). Consistent with our DR-GFP reporter assay, over-expression of all 7 miRNAs in MDA-MB231 cells had a significant impact on RAD51 foci ranging from a 50–75% reduction in foci formation (*Figure 2C*).

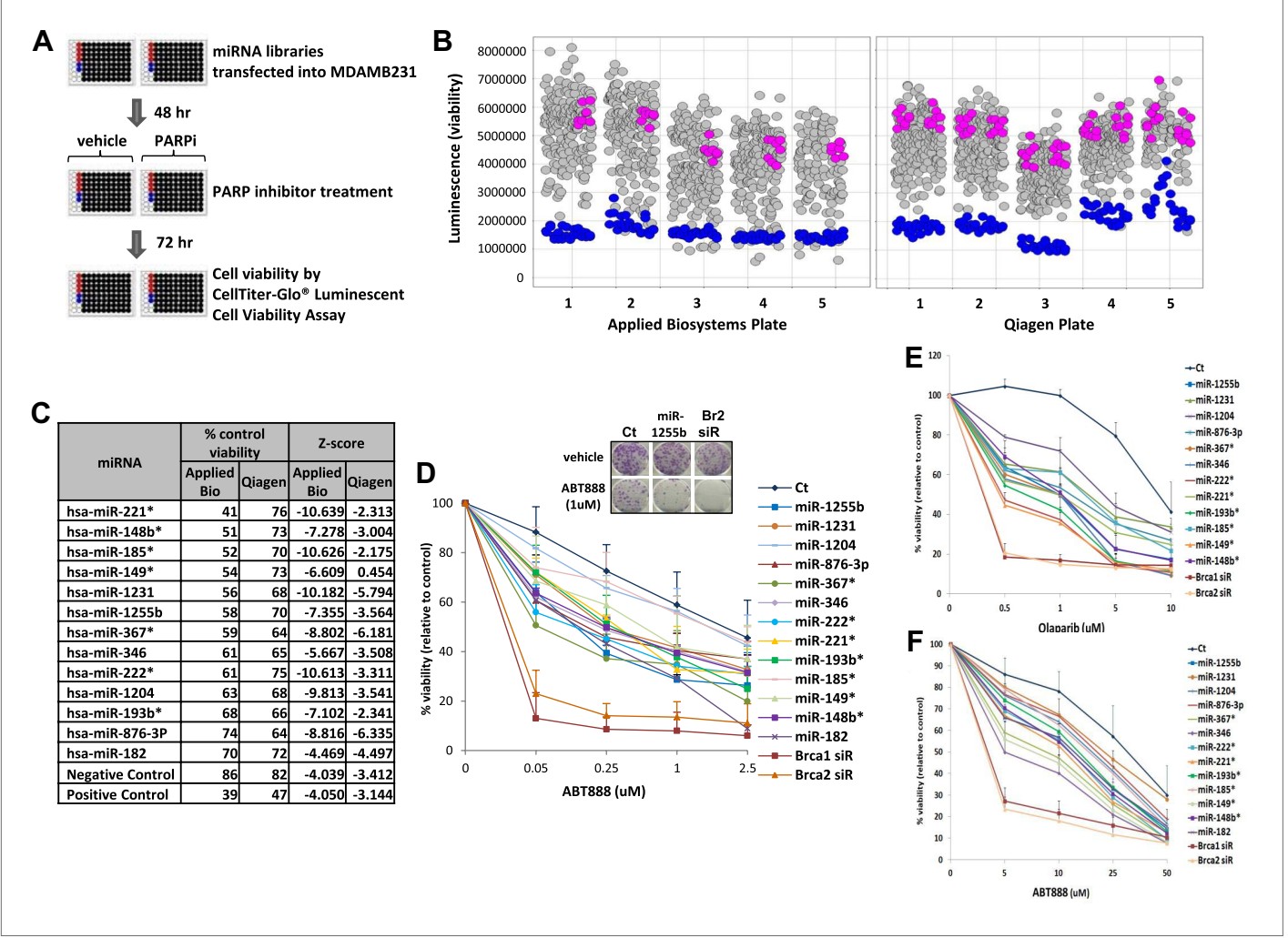

**Figure 1**. miRNA screen for PARP inhibitor sensitivity. (**A**) Schematic of a gain-of-function screen using miRNA mimic libraries from Applied Biosystems and Qiagen to identify miRNAs that sensitize cells to the PARP inhibitor, ABT888. (**B**) Scatter plot (wells/plate) of luminescence (y-axis) as a read-out for viability of each miRNA-transfected well (grey circle) in the presence of ABT888 (20 μM). The plates are numbered in the x-axis. Positive control (BRCA2 siRNA, blue circles) and negative controls (control mimics, pink circles) are shown. Scatter plot for untreated samples is shown in *Figure 1—figure supplement 1*. (**C**) List of top miRNAs from the screen displayed in the order of % control viability along with Z-score. (**D**) Clonogenic survival assay to validate the impact of selected miRNAs on sensitivity to ABT888. MDAMB231 cells were transfected with control miRNA mimics, indicated miRNA mimics, BRCA1 siRNA, or BRCA2 siRNA and treated with vehicle or ABT888, before measuring colony formation. Curves were generated from three independent experiments and a representative image of colony formation with 1 μM ABT888 is shown in the inset. (**E** and **F**) Luminescence-based viability assay was performed in MDAMB231 cells with PARP inhibitor, olaparib (**E**) or in 21NT cells with ABT888 (**F**). Cells were transfected with control miRNA, indicated miRNA mimics, BRCA1 siRNA, or BRCA2 siRNA and treated with vehicle or PARP inhibitor before ATP quantification. Curves were generated from three independent experiments.

The following figure supplements are available for figure 1:

**Figure supplement 1**. miRNAs screen for PARP inhibitor sensitivity.

γ-H2AX formation is a marker for DSBs, and PARP inhibitor treatment of BRCA-deficient cells cause accumulation of γ-H2AX (*Bryant et al., 2005*). To assess whether the miRNAs mirror BRCA deficiency, we assessed γ-H2AX after treating cells with ABT888 (*Figure 2D*). Transfection of mimics for the 7 miRNAs that impacted HR increased γ-H2AX levels 12 hr after treatment with ABT888 and this was maintained through 24 hr. Together, these results strongly suggest that miRNAs (miR-1231, miR-1255b, miR-148b*, miR-876-3p, miR-221*, miR-193b*, and miR-185*), impact the efficacy of DNA repair by impeding the HR-mediated repair of DSBs.

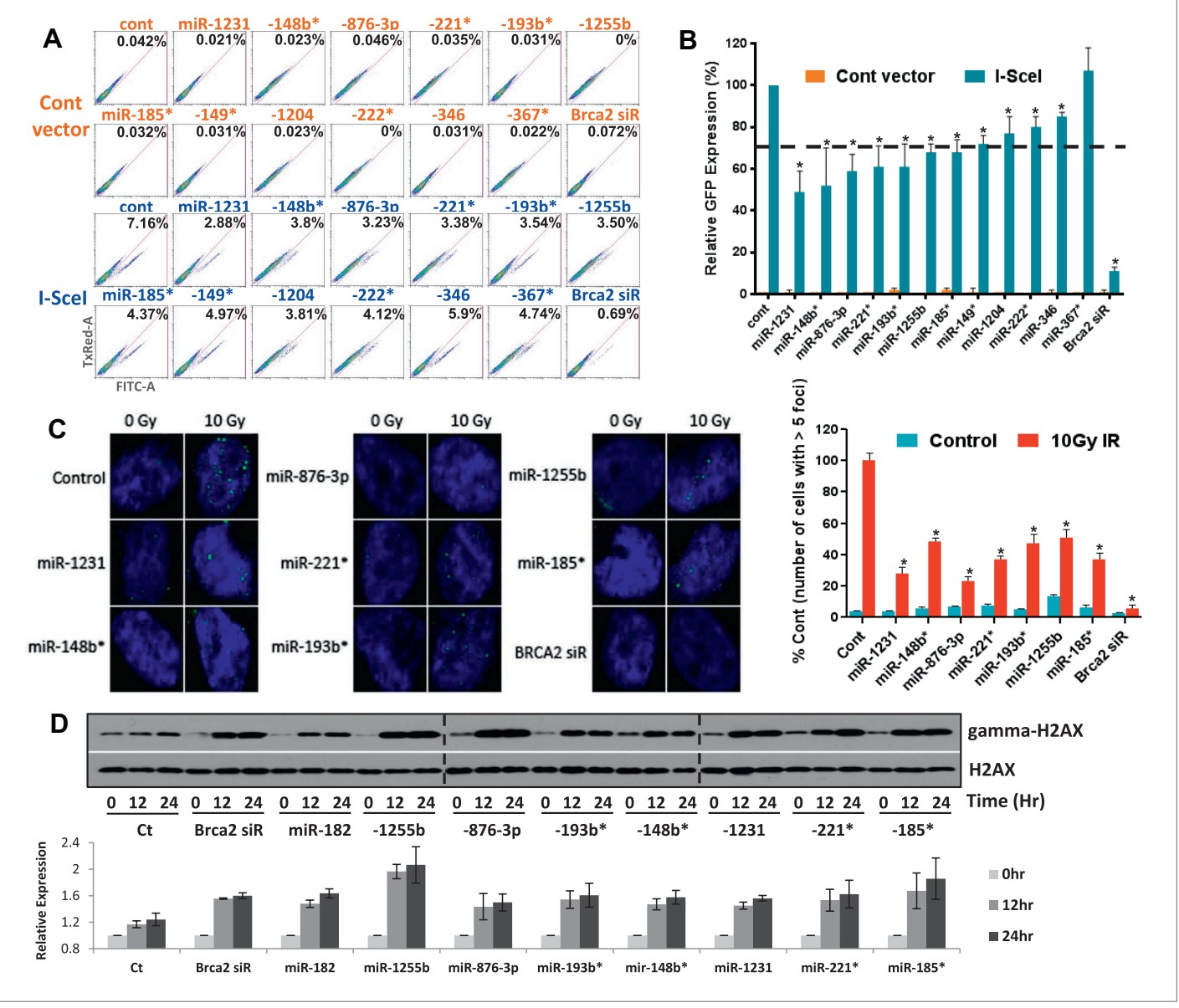

**Figure 2**. miRNAs sensitize cells to PARP inhibitors by targeting HR-mediated DSB repair. (**A** and **B**) Measurement of HR-mediated repair of an I-SceI induced site-specific DSB. Cells carrying a single copy of the recombination substrate (DR-GFP) were transfected with control miRNA mimic, indicated miRNA mimics, or BRCA2 siRNA before transfection with I-SceI or control vector. GFP positive cells were analyzed 48 hr later by flow cytometry (FACS). Representative images of the FACS profile are shown in (**A**), and the mean ± SD of six independent experiments is graphically represented in (**B**). The dotted line represents the cut-off which was set at 70% of the control. (**C**) Analysis of HR-mediated repair by RAD51 focus formation. MDA-MB231 cells were transfected with control miRNA mimic, indicated miRNA mimics, or BRCA2 siRNA, stained for RAD51 (green) and 4′,6-diamidino-2-phenylindole (DAPI) (blue) 6 hr after exposure to IR. The images were captured by fluorescence microscopy and RAD51 focus-positive cells (with >5 foci) were quantified by comparing 100 cells. Mean ± SD of three independent experiments is shown in right panel. * indicates p<0.05. (**D**) γ-H2AX accumulation after treatment with ABT888. Cells were transfected with control miRNA mimic, indicated miRNA mimics, or BRCA2 siRNA and treated with ABT888 (100 µM) before evaluation of γ-H2AX by immunoblotting at indicated time points. Total H2AX served as loading control for these experiments. Images were quantified by ImageJ software and the mean ± SD of three independent experiments is graphically shown.

## Endogenous expression of miRNAs

All our results so far have been derived by artificially introducing miRNAs in cells. To explore the physiological relevance of miR-1231, miR-1255b, miR-148b*, miR-876-3p, miR-221*, miR-193b*, and miR-185*, we first identified cells where these miRNAs are endogenously expressed. From the

perspective of cancer biology, the HR pathway is extremely relevant for breast tumors, specifically triple negative breast cancer (TNBC) (*Lips et al., 2011*), and clinical trials with PARP inhibitors are underway in TNBC patients (*O'Shaughnessy et al., 2009*, *2011*). Therefore, we initially assessed the endogenous expression of the 7 miRNAs in a panel of TNBC lines. Of these seven miRNAs, only miR-1255b, miR-148b*, and miR-193b* had detectable and aberrant over expression in TNBC lines (*Figure 3A*). Therefore, we focused on an in-depth analysis of how miR-1255b, miR-148b*, and miR-193b* impact the HR pathway in TNBC lines and also investigated the physiological relevance of this regulation. It is possible that the other 4 miRNAs (miR-1231, miR-876-3p, miR-221*, and miR-185*) that impact HR may not be expressed in TNBCs but could have physiological relevance in other cell lineages (expression of these miRNAs in other cell types is shown in *Figure 3—figure supplement 1*).

## HR-factors targeted by miR-1255b, miR-148b*, and miR-193b*

We rationalized that the 3 miRNAs identified in the TNBC lines above (miR-1255b, miR-148b*, and miR-193b*) are likely to influence HR by regulating the expression of genes involved in the DNA damage response (DDR). We used a two-pronged approach to identify these genes. First, we used a collection of target prediction algorithms to identify genes targeted by these 3 miRNAs and compared it to the list of DDR proteins (see the 'Materials and methods'). Second, we used a candidate-based approach using the prediction algorithm RNA22 to screen all the genes implicated in DDR for miRNA recognition element of miR-1255b, miR-148b*, and miR-193b*. This gave us a list of 18 proteins which have been previously implicated in HR-mediated repair of DSBs. We then assessed the impact of over-expressing miR-1255b, miR-148b*, and miR-193b* on the mRNA level of these 18 genes (*Figure 3B*). There was at least a 50% reduction in the transcripts of EXO1, MDC1, BRCA1, BRCA2, BRIP1, RAD51, and PALB2 in cells over-expressing either miR-1255b, miR-148b*, or miR-193b*. We next determined the impact of these miRNAs on the protein levels of the putative targets. Over-expressing miR-1255b reduces the cellular levels of BRCA1 and BRCA2, miR-148b* reduces RAD51 and miR-193b* reduces BRCA1, BRCA2, and RAD51 (*Figure 3C*), and this impact is not related to an altered cell cycle (*Figure 3—figure supplement 2*). Although over-expression of miR-1255b, miR-148b*, and miR-193b* correlates with significant increase in the transcripts of certain DNA repair genes (such as MRE11 and NBS1 for miR-1255b and miR-193b*; EXO1 for miR-193b*), there was no detectable increase in protein levels (*Figure 3C*, *Figure 3D*). It is possible that the increase in transcript levels of the DNA repair genes is due to miRNA-mediated down-regulation of transcriptional repressors or mRNA-destablizing proteins, such as AUF1.

## Interaction of miRNAs with BRCA1, BRCA2, and RAD51 transcripts

miRNAs typically suppress gene expression by direct association with target transcripts. To test for association of miR-1255b, miR-148b*, and miR-193b* with their respective target transcripts, we adopted a recently described method for capturing miRNA–mRNA complexes using streptavidin-coated beads from cells transfected with biotinylated forms of the miRNA mimics (*Orom and Lund, 2007*; *Lal et al., 2011*). The amount of BRCA1, BRCA2, and RAD51 transcripts was measured in the pull-downs (GAPDH served as a control transcript) and the enrichment was assessed relative to pull-down with biotinylated control mimic (*Figure 3E–G*). Strikingly, consistent with our previous results, miR-1255b selectively pulled-down BRCA1 and BRCA2 transcripts but not the RAD51 transcript (*Figure 3F*). Conversely, miR-148b* pulled-down RAD51 transcript but not BRCA1 and BRCA2 transcripts, and miR-193b* pulled down all three transcripts. Furthermore, we used a combination of the biotinylated mimics to investigate overlapping targets, combining miR-1255b with miR-193b* and miR-148b*with miR-193b* (*Figure 3G*). Relative to the individual counterparts, the miR-1255b/miR-193b*combination immunoprecipitated increased the amount of BRCA1 and BRCA2 mRNAs and the miR-148b*/miR-193b* combination immunoprecipitated increased RAD51 mRNA (*Figure 3G*).

We next scanned the BRCA1, BRCA2, and RAD51 transcripts for miR-1255b, miR-148b*or miR-193b* binding sites or miRNA recognition element (MRE)s. Canonical MREs are predicted based on the seed rule, that is, the target site within 3' UTR forms Watson–Crick pairs with bases at positions 2 through 7 or 8 of the 5' end of the miRNA (*Lewis et al., 2005*). However, it is noteworthy that there is considerable evidence of 'seedless' or non-canonical association of miRNAs with target transcripts including sites in the coding sequence (CDS) (*Didiano and Hobert, 2006*; *Grimson et al., 2007*; *Chi et al., 2009*; *Lal et al., 2009a*; *Hafner et al., 2010*; *Shin et al., 2010*; *Chi et al., 2012*; *Loeb et al., 2012*).

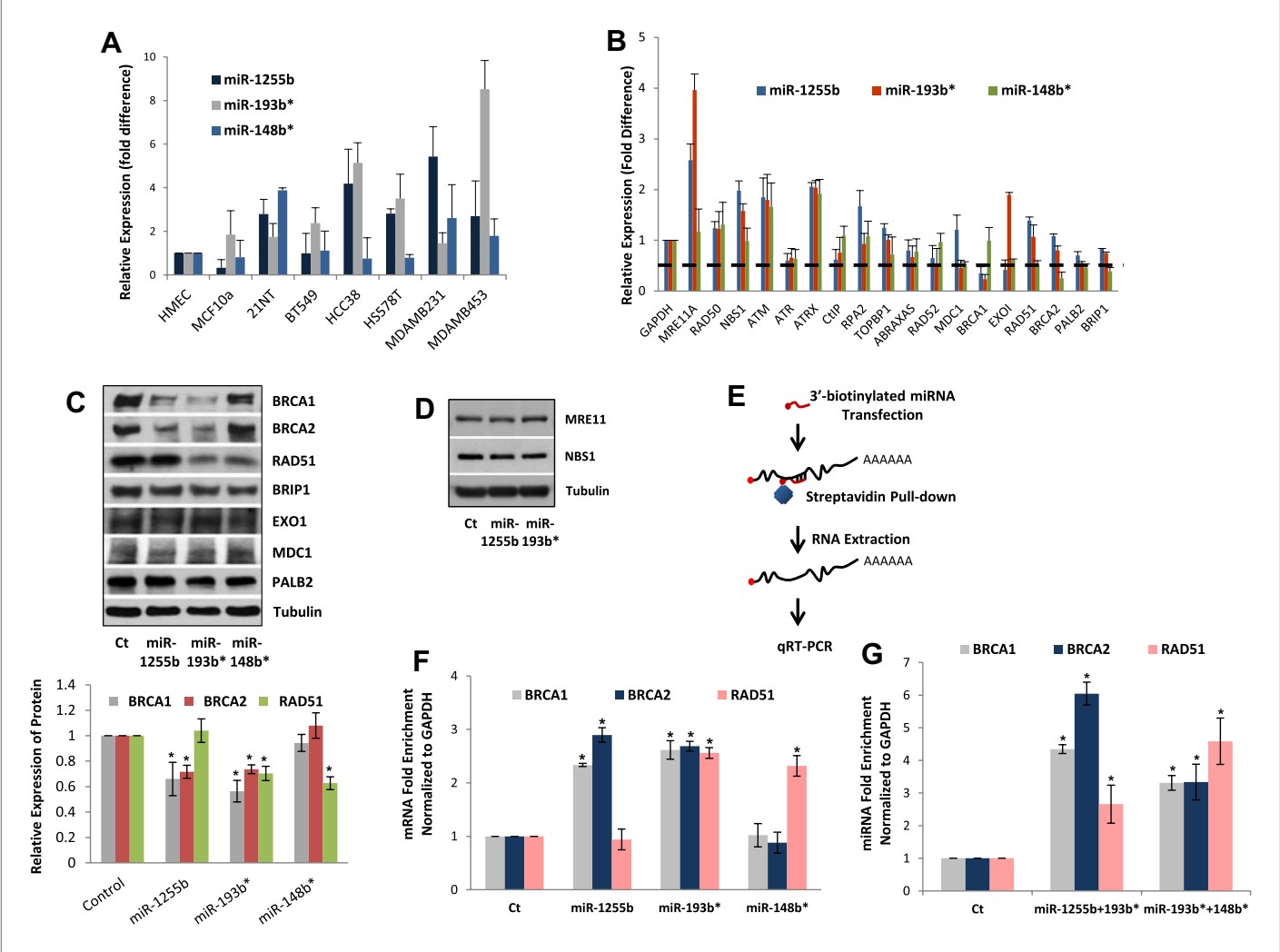

**Figure 3**. miR-1255b, miR-193b*, and miR-148b* regulate PARP inhibitor sensitivity by regulating expression of HR factors in TNBCs. (**A**) miRNA expression profile in a panel of breast cancer lines. Endogenous expression of indicated miRNAs was quantified by qRT-PCR (normalized to 5srRNA) and represented relative to non-tumorigenic breast epithelial cell, HMEC. Expression of miR-1255b, miR-193b*, and miR-148b* were detected in these lines. Mean ± SD of four independent experiments is shown. (**B–D**) Expression of DDR genes is impacted by miR-1255b, miR-193b*, and miR-148b*. MDA-MB231 cells were transfected with control mimic or mimics for miR-1255b, miR-193b*, and miR-148b* and mRNA levels of predicted and prioritized DDR genes were analyzed by qRT-PCR using gene-specific primers and normalized to GAPDH. Mean ± SD of three independent experiments is shown (**B**). (**C** and **D**) Cell lysates were then analyzed by immunoblot for factors which had ≥50% reduction in mRNA in cells transfected with miR-1255b, miR-193b*, and miR-148b*. Images were quantified by ImageJ software and the mean ± SD of three independent experiments is graphically shown, * indicates p<0.05. (**E–G**) Interaction of target transcripts with miR-1255b, miR-193b*, and miR-148b*. MDA-MB231 cells were transfected with biotinylated-control mimics or biotinylated mimics for miR-1255b, miR-193b*, and miR-148b* as a single (**F**) or a combination (**G**). The immunoprecipitated RNA was analyzed by qRT-PCR using gene-specific primers and normalized to GAPDH. Mean ± SD of three independent experiments is shown and statistical significance of enrichment of specific gene transcripts is indicated by * (p<0.05). The principle steps of the method are illustrated in *Figure 3E*.

The following figure supplements are available for figure 3:

**Figure supplement 1**. Expression of the excluded miRNAs.

**Figure supplement 2**. The effect of miRNAs on cell cycle.

None of the three transcripts BRCA1, BRCA2, and RAD51 had MREs with a canonical seed region in the 3'UTR for miR-1255b, miR-148b*, and miR-193b*. We have previously used the PITA algorithm to expand the search criterion and identify non-canonical MREs for specific miRNA/mRNA combinations (*Lal et al., 2009a*). The PITA algorithm identifies base pairing beyond the 5'end of the

miRNA, allows G:U wobbles or seed mismatches and predicts MREs across the entire target transcript, not just the 3'UTR. Using this approach, we observed that consistent with our experimental data, there were putative non-canonical MREs for miR-193b*, miR-1255b, and miR-148b* in the transcripts of BRCA1, BRCA2, and RAD51 (*Figure 4A*). Most of these sites had moderate to low evolutionary conservation (*Figure 4—figure supplement 1*).

To verify further that BRCA1, BRCA2, and RAD51 are targets of miR-1255b, miR-148b*, and miR-193b* and to confirm that the interaction is mediated by the predicted MREs, we used the luciferase reporter assay which is a surrogate for target protein. The MREs were cloned in the 3'UTR of the luciferase gene, and expression monitored in cells transfected with mimics for miR-1255b, miR-193b*, and miR-148b*(*Figure 4A,B*). As anticipated, there was significant decrease in luciferase activity, and this was 'rescued' by point mutations that disrupt base pairing between miR-1255b, miR-193b*, and miR-148b* and their corresponding MREs in BRCA1, BRCA2, and RAD51 (*Figure 4A,B*). Analyzing all the MREs individually, we compared the relative impact of each MRE on luciferase activity (*Figure 4C*). To confirm the interaction of endogenous miR-1255b, miR-193b*, and miR-148b* with specific MREs in the BRCA1, BRCA2, and RAD51 transcripts, we adopted a loss-of-function approach. We used miRNA inhibitors (also known as antagomirs, ANTs) that are single-stranded chemically enhanced oligonucleotides designed to irreversibly bind endogeneous miR-1255b, miR-193b* and miR-148b and suppress their activity. We estimated luciferase activity after inhibiting the miRNAs using antagomirs and, consistent with our previous results, found that inhibition of miR-1255b enhanced luciferase activity of the BRCA1 and BRCA2 reporter construct, inhibition of miR-148b* enhanced luciferase activity of the RAD51 reporter construct, and inhibition of miR-193b* enhanced luciferase activity of the BRCA1, BRCA2, and RAD51 reporter constructs (*Figure 4D*). The specificity of the MREs was further validated as the mutant versions of the luciferase reporters were immune to the antagomirs (*Figure 4D*).

The luciferase reporter assays with MREs provide important information regarding the miRNA/mRNA association but have limited physiological relevance. To determine the functional significance of non-canonical MREs in the BRCA1, BRCA2, and RAD51 transcripts we generated expression constructs without the MREs by either deletion (MREs in 3'UTR) or mutation (MREs in CDS) of them. Next, MDA-MB231 cells were co-transfected with (i) miR-1255b and BRCA1 or BRCA2 expression plasmid lacking miR-1255b binding sites; (ii) miR-193b* and BRCA1 or BRCA2 or RAD51 expression plasmid lacking miR-193b* binding sites; (iii) miR-148b* and a RAD51 expression plasmid lacking miR-148b* binding sites. First, the BRCA1, BRCA2, and RAD51 expression constructs lacking the specific MREs completely restored the expression of these genes in the presence of the corresponding miRNA mimic further validating the predicted MREs (*Figure 5A*, lower panel). Furthermore, in regard to ABT888 sensitivity, expression of BRCA1 or BRCA2 significantly 'rescued' the impact of miR-1255b, expression of BRCA1 or BRCA2 or RAD51 significantly 'rescued' the impact of miR-193b*, and expression of RAD51 significantly 'rescued' the impact of miR-148b* (*Figure 5A*, upper panel). Together, these results strongly suggest that miR-1255b, miR-193b*, and miR-148b* influence HR-mediated repair of DSBs and PARP inhibitor sensitivity by regulating expression of BRCA1, BRCA2, and RAD51.

## Impeding miRNAs cause ectopic expression of HR factors in G1 cells

Ectopic over-expression of miR-1255b, miR-193b*, and miR-148b* in cells down-regulates HR factors and influences PARP inhibitor sensitivity. However, the cellular function of endogenously expressed miR-1255b, miR-193b*, and miR-148b* remains unclear. To address this issue, we quantified their expression in cycling cells, specifically during the G1 to S transition. Consistent with previous reports (*Gudas et al., 1996*; *Vaughn et al., 1996*; *Chen et al., 1997*), we observe that mRNA levels of BRCA1, BRCA2, and RAD51 are enhanced in the S-phase (*Figure 5B,C*). miR-1255b, miR-193b*, and miR-148b* inversely correlate with their target transcripts, and are significantly down-regulated as cells move into the S-phase. This was observed both in the breast tumor line, MDA-MB231 (*Figure 5B*) and the non-tumorigenic breast epithelial line MCF10A (*Figure 5C*). Antagonizing these miRNAs induces a specific increase in target transcripts (*Figure 5D*) and proteins (*Figure 5E*) in the G1-phase, that is transfection of the miR-1255b ANT causes significant increase in BRCA1 and BRCA2 and not RAD51, miR-148b* ANT increases RAD51 and miR-193b* ANT increases BRCA1, BRCA2, and RAD51 in G1 cells (*Figure 5D,E*, cell cycle profiles are shown in *Figure 5—figure supplement 1*). Combined inhibition of miRNAs with overlapping targets causes a synergistic increase in target transcripts in G1 (*Figure 5F*). There is a significant increase in BRCA1 and BRCA2 transcripts in cells co-transfected with miR-1255b ANT and miR-193b*ANT, and a significant increase in RAD51 mRNA in cells co-transfected with

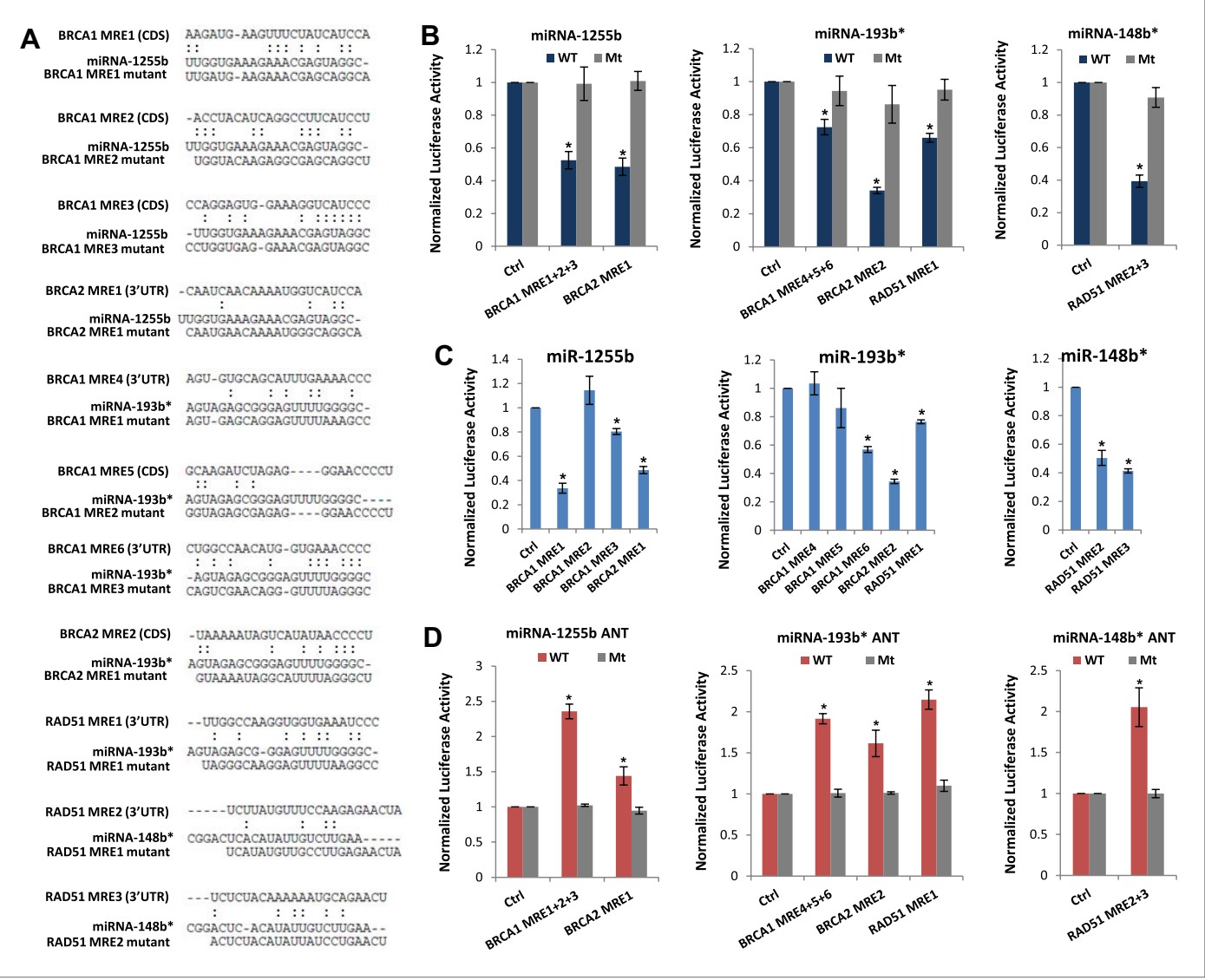

**Figure 4**. Predicted miRNA recognition sites (MREs) of miRNAs and their impact on targets. (**A**) Predicted MREs were obtained from PITA (http://genie. weizmann.ac.il/pubs/mir07/mir07_prediction.html) and their mutants were generated by mutating nucleotides providing complementarity and G-U wobble to corresponding miRNAs. The region where MRE is located in the gene is indicated in the parentheses. CDS: coding sequence, 3'UTR: 3' untraslated region. (**B**) Luciferase reporter assay to assess direct interaction of miR-1255b, miR-193b*, and miR-148b* with BRCA1, BRCA2, and RAD51. Combinations of predicted miRNA recognition sites (MREs) for each putative target transcript of miR-1255b, miR-193b*, and miR-148b* were cloned into the luciferase reporter vector and transfected in MDA-MB231 cells along with the indicated miRNA mimics. *Renilla* luciferase activity of the reporter was measured 48 hr after transfection by normalization to an internal *firefly* luciferase control. Mean ± SD of three independent experiments is shown and statistical significance is indicated by * (p<0.05). (**C**) Luciferase reporter assay for individual MREs for each target of miRNAs was performed in the same way as described in *Figure 4B*. Mean ± SD of three independent experiments is shown and statistical significance is indicated by *(p<0.05). (**D**) Luciferase reporter assay with miR-1255b, miR-193b*, and miR-148b* ANTs. Combinations of predicted miRNA recognition sites (MREs) in the luciferase vector for each putative target transcript of miR-1255b, miR-193b*, and miR-148b* were transfected in MDA-MB231 cells along with the indicated miRNA ANTs. *Renilla* luciferase activity of the reporter was measured 48 hr after transfection by normalization to an internal *firefly* luciferase control. Mean ± SD of three independent experiments is shown and statistical significance is indicated by *(p<0.05).

The following figure supplements are available for figure 4:

**Figure supplement 1**. Conservation of predicted miRNA recognition sites (MREs) of miRNAs.

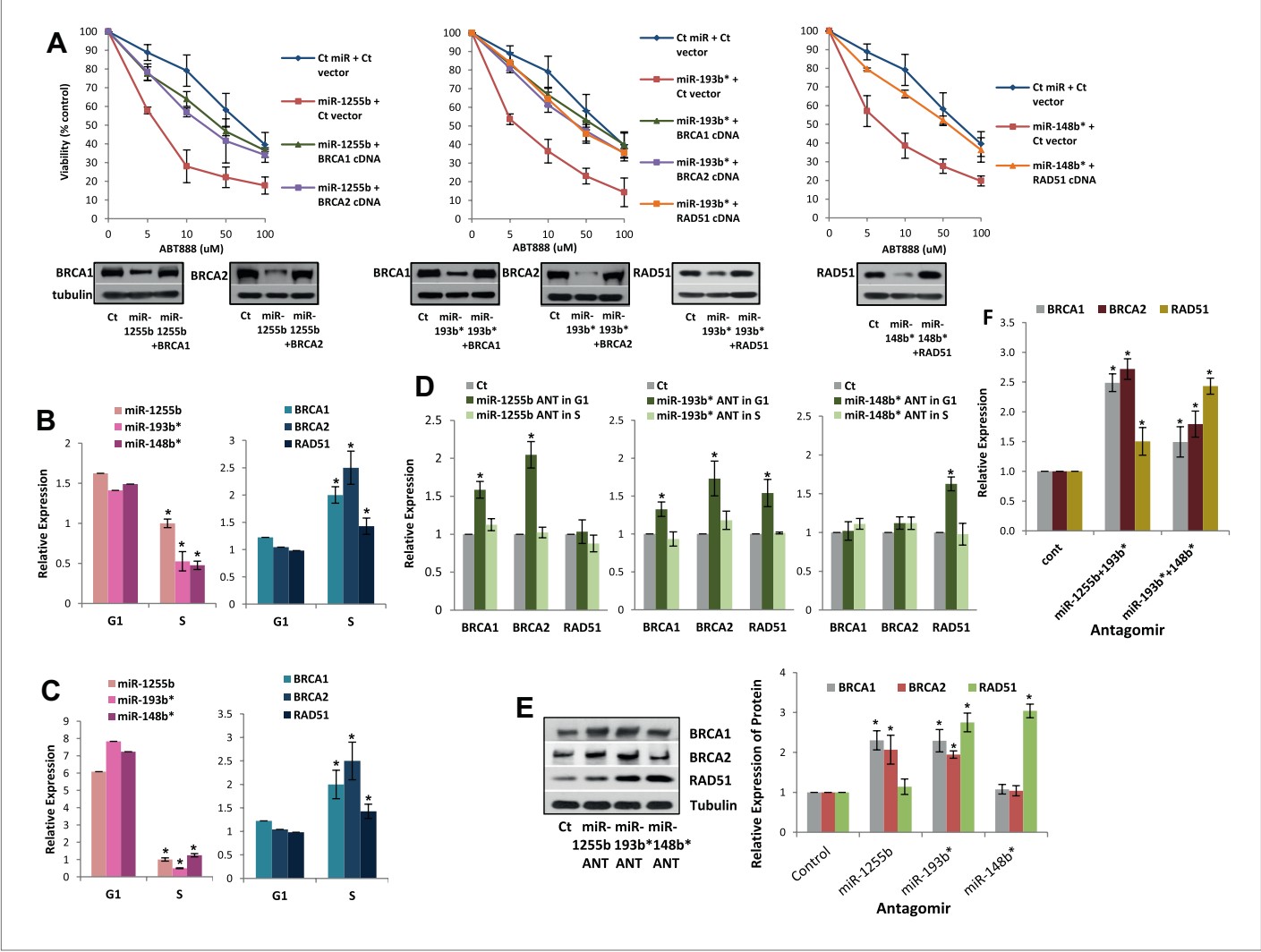

**Figure 5**. Impact of miRNAs on DSB repair in different phases of the cell cycle. (**A**) Rescue of the impact of miRNAs on ABT888 sensitivity. MDAMB231 cells were transfected with control miRNA or indicated miRNA mimics with or without target gene cDNAs (lacking MREs) and treated with vehicle or ABT888, before viability assay by ATP quantification. Expression of each target protein is examined by immune blot. (**B** and **C**) Expression of miRNAs and target transcripts in synchronized cells. MDAMB231 (**B**) or MCF10A (**C**) cells were synchronized with mimosine and the relative amount of miR-1255b, miR-193b*, and miR-148b* or BRCA1, BRCA2, and RAD51 mRNA for G1- or S-phase was determined by qRT-PCR (normalized to RNU1 or GAPDH, respectively). Mean ± SD of three independent experiments is shown and statistical significance is indicated by *(p<0.05). (**D**–**F**) Impact of inhibiting miRNAs on targets in G1 cells. MDAMB231 cells were transfected with control ANT or ANTs for miR-1255b, miR-193b*, and miR-148b* as a single (**D**) or a combination (**F**). Subsequently, the cells were synchronized with mimosine and BRCA1, BRCA2, and RAD51 mRNA was assessed by qRT-PCR (normalized to GAPDH) in the G1 and/or S-phase (**D** and **F**). Cell lysates from G1 cells were analyzed by immunblot for BRCA1, BRCA2, and RAD51 (**E**). Images were quantified by ImageJ software and the mean ± SD of three independent experiments is shown, * indicates p<0.05.

The following figure supplements are available for figure 5:

**Figure supplement 1**. The impact of miRNA antagomirs (ANTs) on cell cycle progression.

miR-148b*ANT and miR-193b*ANT. These results suggest that in proliferating cells miRNAs suppress HR in the G1 phase.

## miRNA-mediated suppression of HR proteins in G1 facilitates DSB repair

There is emerging evidence that loss of NHEJ factors allows the initiation of HR in G1 cells and leads to the persistence of unrepaired DSBs (*Helmink et al., 2011*; *Escribano-Diaz et al., 2013*). Therefore, it is feasible that up-regulation of HR factors due to the inhibition of miR-1255b, miR-193b*, and

miR-148b* in G1 cells may also de-stabilize the balance of the HR and NHEJ pathways and thereby impede DSB repair. To test this hypothesis, asynchronous cells were transfected with antagomirs for miR-1255b, miR-193b* and miR-148b* and stained with Cyclin A to visualize S/G2 cells and distinguish them from G1 cells; γ-H2AX was utilized as a marker for DSBs. Inhibition of miR-1255b and miR-193b* causes a moderate but statistically significant increase in basal γ-H2AX specifically in G1 cells (*Figure 6—figure supplement 1A*). The impact of inhibiting the miRNAs on DSB repair in G1 cells is even more pronounced when these cells are exposed to IR (*Figure 6A,B*). BRCA1 promotes

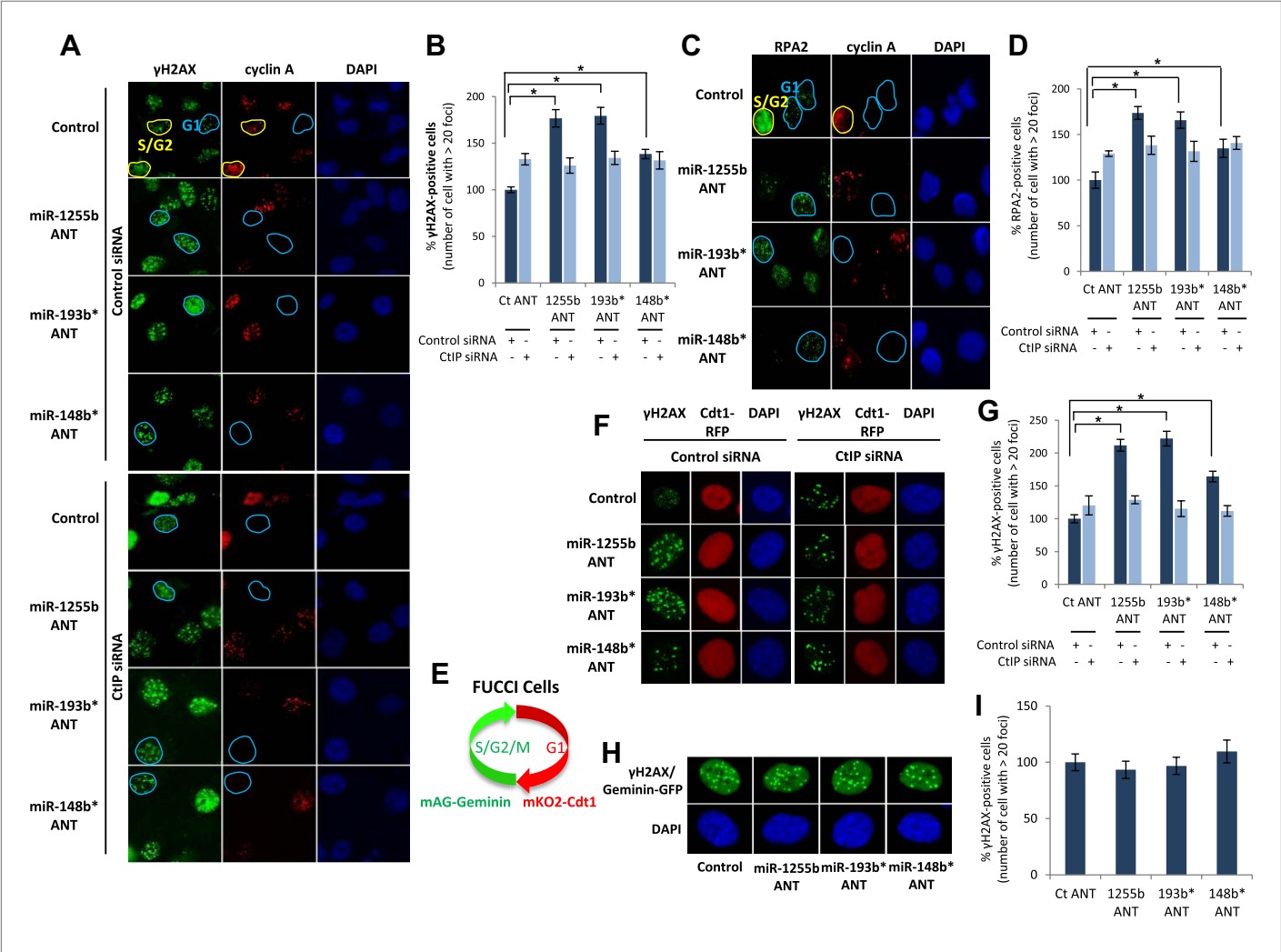

**Figure 6**. Impact of inhibiting miRNAs on DSB repair. (**A–D**) Impact of inhibiting miRNAs on DSB repair in the G1 phase of MDA-MB231 cells. Cells were transfected with control ANT or ANTs for miR-1255b, miR-193b*, and miR-148b* with or without 20 nM CtIP siRNA, exposed to IR (5 Gy) and stained for γ-H2AX (green) (**A**) or RPA2 (green) (**C**), cyclin A (red) and 4′,6-diamidino-2-phenylindole (DAPI) (blue). The images were captured by fluorescence microscopy and the number of γ-H2AX foci (**A**) or RPA2 foci (**C**) was calculated from 100 cells. Mean ± SD of three independent experiments is graphically represented (**B** and **D**). * indicates p<0.05. (**E–I**) Impact of inhibiting miRNAs on DSB repair in different phase of RPE-1 cells. RPE-1 cells expressing the Fucci system (illustrated in *Figure 6E*) were transfected with control ANT or ANTs for miR-1255b, miR-193b*, and miR-148b* with or without 20 nM CtIP siRNA, exposed to IR (5 Gy) and stained for γ-H2AX (green) and 4′,6-diamidino-2-phenylindole (DAPI) (blue). The images were captured by fluorescence microscopy and the number of γ-H2AX foci in G1 cells (red, mKO2-Cdt1) was calculated from 100 cells (**F** and **G**). The images were captured by fluorescence microscopy and the number of γ-H2AX foci (green foci) in S/G2/M cells (green background, mAG-Geminin) was calculated from 100 cells (**H**). Mean ± SD of three independent experiments is graphically represented (**I**). * indicates p<0.05.

The following figure supplements are available for figure 6:

**Figure supplement 1**. The impact of miRNAs on DNA repair during cell cycle.

end resection at DSBs via its interaction with CtIP (*Yun and Hiom, 2009*) and that is the key step in blocking NHEJ. The significant increase in residual DSBs in cells transfected with antagomirs for miR-1255b, miR-193b* and miR-148b* is 'rescued' by silencing CtIP (*Figure 6A,B*; silencing efficacy shown in *Figure 6—figure supplement 1B*), suggesting that increased levels of BRCA1 by miRNA inhibition in G1 promotes CtIP-mediated resection.

RPA2 foci mark the presence of single-strand DNA (ssDNA), a consequence of end resection. Consistent with resection of DNA ends, and production of ssDNA, we observe a significant increase in RPA2 foci in the G1 phase of cells transfected with antagomirs for miR-1255b, miR-193b* and miR-148b* and again the depletion of CtIP negates the formation of RPA2 foci in these cells (*Figure 6C,D*). To further confirm the cell cycle phase specificity of this phenotype using a different approach and a diploid cell line with relatively few genomic abnormalities, we utilized the Fucci system (*Sakaue-Sawano et al., 2008*) to visualize the G1 phase (mKO2-Cdt1, red fluoresence) in hTERT-immortalized retinal pigment epithelial cell line (RPE-1) cells (*Figure 6E–G*). Fucci (*f*luorescent *u*biquitination-based *c*ell-*c*ycle *i*ndicator) utilizes cell cycle-dependent degradation of Cdt1 and Geminin to mark G1 and S/$G_2$/M cells by fusing the red (mKO2) and green (mAG) fluorescent proteins to Cdt1 and Geminin, respectively (illustrated in *Figure 6E*). Consistent with the previous results, inhibition of miR-1255b, miR-193b*, and miR-148b* in RPE-1 causes a CtIP-dependent increase in residual DSBs in G1 cells (*Figure 6F,G*), but this effect is not observed in the S-phase (*Figure 6H,I*). Together, these results strongly suggest that miR-1255b, miR-193b*, and miR-148b* play a role in preventing the initiation of deleterious HR-mediated DSB repair in the G1 phase.

## Deletion of miRNAs correlate with increased LOH in primary ovarian tumors

The two major mechanisms proposed to cause LOH are deletion of chromosomal fragments/chromosome loss and mitotic recombination between homologous alleles (*Lasko et al., 1991*). Mitotic recombination is a consequence of HR-mediated repair of DSBs in the G1 phase. Recent studies demonstrate that homologous chromosomes are in close proximity at DSB sites in the G1 phase (*Gandhi et al., 2012*, *2013*) further supporting the notion that HR-mediated DSB repair is feasible in G1 cells and this in turn could lead to LOH. Based on our experimental data, the prediction would be that loss of miR-1255b, miR-193b*, and miR-148b* would correlate with enhanced LOH events that are occurring due to mitotic recombination. To test this hypothesis, we utilized the data from the *Cancer Genome Atlas Research Network, 2011* (TCGA, [2011]), where the processed LOH data for high-grade serous ovarian tumors is readily available. The HR-pathway and expression of BRCA1 and BRCA2 is extremely relevant in ovarian tumors, and we observed that there is detectable expression of miR-1255b, miR-148b*, and miR-193b* in a panel of ovarian tumor lines (*Figure 7A*). The increased γ-H2AX foci correlating with inhibition of miR-1255b, miR-193b*, and miR-148b* is not observed in a BRCA1-mutant ovarian cell line UWB1.289 (*Figure 7—figure supplement 1*) confirming the importance of BRCA1 in the phenotype induced by these miRNAs.

We compared the number of LOH events for a cohort (417 tumors) of high-grade serous ovarian tumors that have deletions of miR-1255b, miR-148b*, and miR-193b* and those that are copy neutral at those loci. We observed that the number of LOH events is significantly higher in tumors with deletions of miR-148b* and miR-193b* relative to the tumors that are diploid copy neutral at these loci. The number of LOH events was also moderately higher (One tailed Mann Whitney's U test; p<0.08) in tumors with deletion of miR-1225b compared to the copy neutral ones (*Figure 7B*). It is noteworthy that mature miR-1255b is encoded from two distinct genomic loci (miR-1255b-1 on chr4: 36427988-36428050 and miR-1255b-2 on chr1: 167967898-167967964). In our deletion analysis, we correlated LOH events with deletion of either miR-1255b-1 or miR-1255b-2, therefore the moderate correlation of LOH with loss of miR-1255b may be due to the presence of the second copy. The limited sample size with deletions of both miR-1255b-1 and miR-1255b-2 precludes any statistical analysis. Reflecting overall genomic instability, there is a significant increase in somatic amplifications and deletions in tumors with deletions in miR-1255b, miR-148b*, and miR-193b* (*Figure 7C*).

To validate our analysis in a second cohort, we obtained processed LOH and somatic copy number alteration data for 47 high-grade serous ovarian tumors, which had been processed at Dana-Farber/Harvard Cancer Center (DF/HCC). The details of these samples have been previously described (*Wang et al., 2012b*). We compared the number of copy neutral LOH events between the samples that have deletions of miR-1255b, miR-148b*, and miR-193b*and those that are copy neutral at those

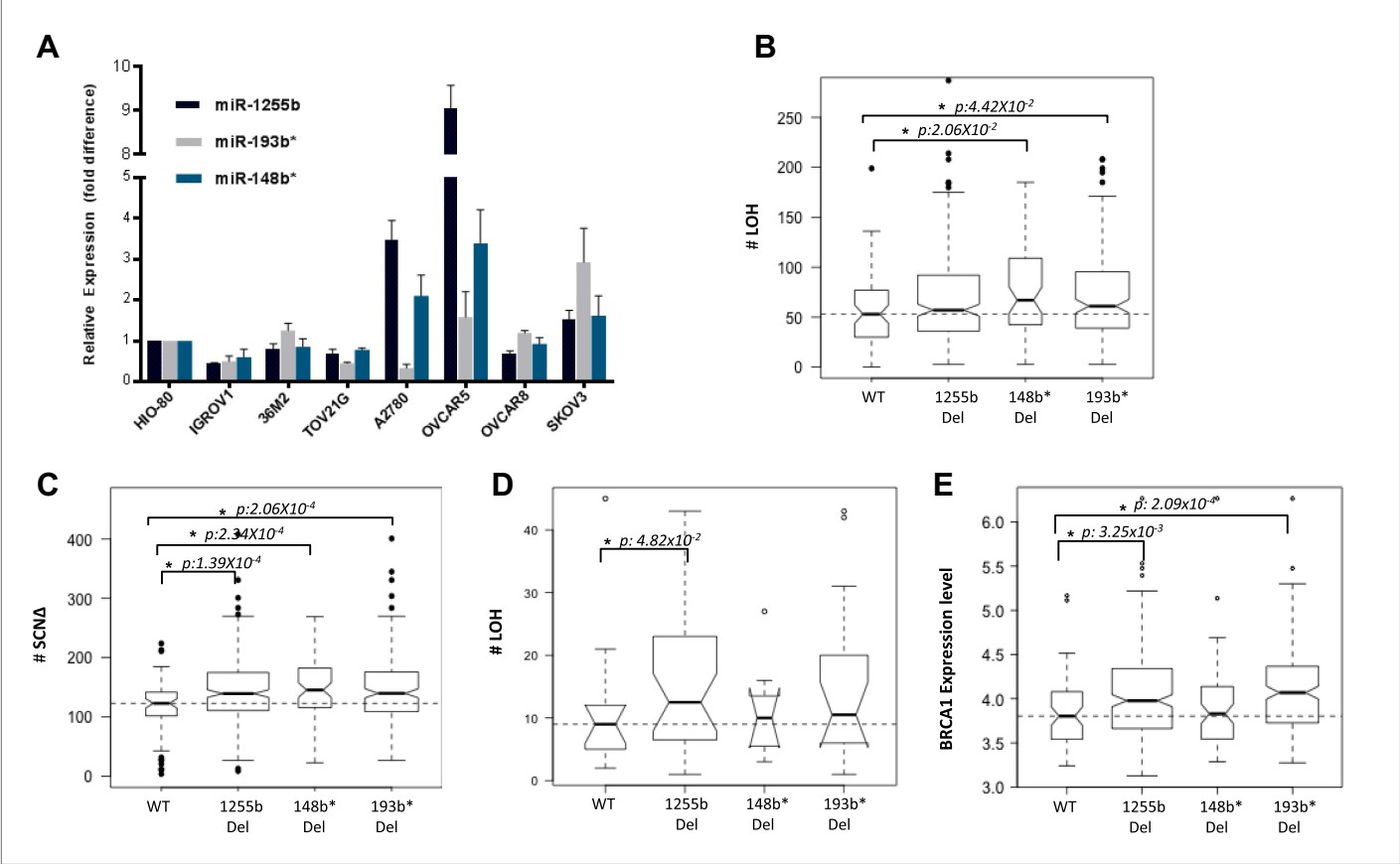

**Figure 7**. Correlation of LOH with loss of miRNAs. (**A**) miRNA expression profile in a panel of ovarian cancer lines. Endogenous expression of indicated miRNAs was quantified by qRT-PCR (normalized to 5srRNA) and represented relative to non-tumorigenic ovarian epithelial cell, HIO-80. Expression of miR-1255b, miR-193b*, and miR-148b* were detected in these lines. (**B** and **C**) Correlation of LOH with deletion of miRNAs in TCGA data set. Box plots show the frequency of (**B**) LOH or (**C**) somatic copy number amplification or deletion (SCNΔ) in the 418 high-grade serous ovarian tumors from TCGA that have no amplifications or deletions of any of these 3 miRNAs (WT), against those with deletion of 1255b (either −1 or −2), 148b* or 193b*. The LOH events are >1 Mb. (**D**) Correlation of LOH with deletion of miRNAs in DF/HCC data set. Box plot shows the frequency of LOH in 47 high-grade serous ovarian tumors that have no amplifications or deletions of any of these 3 miRNAs (WT), against those with deletion of miR-1255b (either 1 or 2), miR-148b* or miR-193b*. The LOH events are >1 Mb. (**E**) Correlation of BRCA1 expression with deletion of miRNAs in TCGA data set. Box plot shows expression levels of BRCA1 in the 418 high-grade serous ovarian tumors from TCGA that have no amplifications or deletions of any of these 3 miRNAs (WT), against those with deletion of miR-1255b (either 1 or 2), miR-148b* or miR-193b*. In (**B**–**E**), statistical significance was calculated using one tailed Mann Whitney's U test and significant differences (p<0.05) indicated with asterisk. Width of the bars indicates the number of samples. The horizontal line indicates the median of the WT set.

The following figure supplements are available for figure 7:

**Figure supplement 1**. miRNA dependent regulation of DSB repair during cell cycle.

loci (*Figure 7D*). Consistent with the analysis of the TCGA data, the number of LOH events is higher in the tumors with deletions of miR-1255b, miR-148b*, and miR-193b* relative to the samples that are diploid copy neutral at these loci. The statistical significance was limited due to small size of the data set, but the patterns were consistent even with alternative cutoffs. In our analysis, we compared the copy neutral LOH events to specifically focus on LOH events that are likely to occur due to recombination between homologous alleles. Finally, in agreement with our experimental data, there is a significant increase in BRCA1 expression in the tumors with deletions in miR-1255b and miR-193b*(*Figure 7E*), which we speculate is the underlying reason for the increase in LOH and overall genomic instability.

## Discussion

Contrary to other cellular processes and signaling pathways, there has been limited investigation of miRNAs in DSB repair (*Wan et al., 2011*; *Chowdhury et al., 2013*). In the few recent studies

(*Hu et al., 2010*; *Yan et al., 2010*; *Wang et al., 2011*, *2012a*; *Dimitrov et al., 2013*; *Huang et al., 2013*; *Neijenhuis et al., 2013*), including ones from our group (*Lal et al., 2009b*; *Moskwa et al., 2011*), the focus has been the impact of a single miRNA on an individual DSB protein. A key conceptual limitation to this approach is that miRNAs typically target hundreds of transcripts and are more likely to have a profound impact on a pathway like the HR repair pathway by regulating several HR factors. Here, we systematically identified miRNAs influencing HR (miR-1231, miR-1255b, miR-148b*, miR-876-3p, miR-221*, miR-193b*, and miR-185*) and observed that three of these miR-NAs (miR-1255b, miR-148b*, and miR-193b*) regulate expression of BRCA1, BRCA2, and RAD51. Intriguingly, they lack canonical binding sites in the 3'UTR of BRCA1, BRCA2, and RAD51, but interact with these transcripts via non-canonical MREs. These results underline the importance of binding of miRNAs to their target transcripts that are not restricted to base pairing at 'seed regions', and highlight the shortcomings of current prediction algorithms in identifying functional targets of miRNAs.

The choice of DSB repair pathway is critical for maintaining genomic stability (*Chapman et al., 2012b*). Specifically 53BP1 and its associated factors (RIF1 and PTIP) promote NHEJ and counter BRCA1 thereby balancing the process of DSB repair between HR and NHEJ (*Bouwman et al., 2010*; *Bunting et al., 2010*; *Chapman et al., 2012a*; *Callen et al., 2013*; *Di Virgilio et al., 2013*; *Escribano-Diaz et al., 2013*; *Zimmermann et al., 2013*). Loss of the 53BP1-associated protein RIF1 or the protein H2AX allows the BRCA1/CtIP-mediated resection of DNA ends in G1 cells impeding NHEJ and leading to the persistence of DNA lesions (*Helmink et al., 2011*; *Escribano-Diaz et al., 2013*). Conversely, it is likely that the ectopic increase in expression of BRCA1 and other HR factors in the G1 phase will also disrupt the balance of DSB repair. Our results represent an intriguing example of an organized, miRNA-driven physiological system for controlling the level of endogenous HR factors in the G1 phase of the cell cycle. Down-regulating the expression of HR factors, particularly, an HR-initiating factor like BRCA1 in the G1 cells is necessary for maintaining genomic stability. Inhibiting miR-1255b and miR-193b* allows increased BRCA1 expression in G1 cells leading to CtIP-mediated resection and a potential block in NHEJ. This will disrupt the balance of HR and NHEJ (*Figure 8*) with significant increase in genomic instability. Furthermore, execution of HR-mediated repair in G1 cells would result in LOH which has the potential of revealing recessive oncogenic mutations. The loss of miR-1255b, miR-148b*, and miR-193b* largely correlates with increased genomic instability and a higher frequency of LOH, which is consistent with our experimental data. In some cases, the correlations are moderate and not statistically significant, and this may reflect the heterogeneity of primary tumors, limited sample size and the technical shortcomings of large sets of genomic data. These shortcomings notwithstanding, there is a clear pattern that emerges from our experimental data and clinical analysis that strongly suggests that miRNAs regulate the optimal expression of HR proteins in the course of the cell cycle and prevent ectopic activation of the HR pathway. A single copy loss of BRCA1 (*Konishi et al., 2011*), BRCA2 (*Popova et al., 2012*), or RAD51 (*Smeenk et al., 2010*) impacts genomic stability giving credence to the idea that a miRNA-mediated alteration in the cellular levels of these proteins may significantly impact the DNA repair process. Furthermore, for miR-1255b and miR-193b*, there is a moderate reduction in multiple HR factors, which further compounds their impact on the HR repair pathway. The underlying mechanism of cell cycle-dependent expression of miRNAs regulating HR factors remains to be investigated in future studies.

## Materials and methods

### Constructs

BRCA1 MRE1 and BRCA2 MRE2 located in the gene coding sequence were mutated by site-directed mutagenesis. The primers used are as follows:

BRCA1 MRE1-1-US, AAGTTTCTATCAGGCAAAGTATGGGCT, BR1_Res1-1-DS, AGCCCATACTT TGCCTGATAGAAACTT; BR1_Res1-2-US, CTAAAAGATGAAGAAACTATCAGGCAAA, BR1_Res1-2-DS, TTTGCCTGATAGTTTCTTCATCTTTTAG; BR2_Res2-US, AAATAGTCATATAAGGGCTCAGATGTTATTT, BR2_Res2-DS, AAATAACATCTGAGCCCTTATATGACTATTT.

### miRNA screen

miRNA screen was carried out in triplicates using miRNA mimic libraries from Applied Biosystems (Pre-miR miRNA Precursor Library, 885 miRNAs) and Qiagen (miScript miRNA Mimic Library, 875 miRNAs),

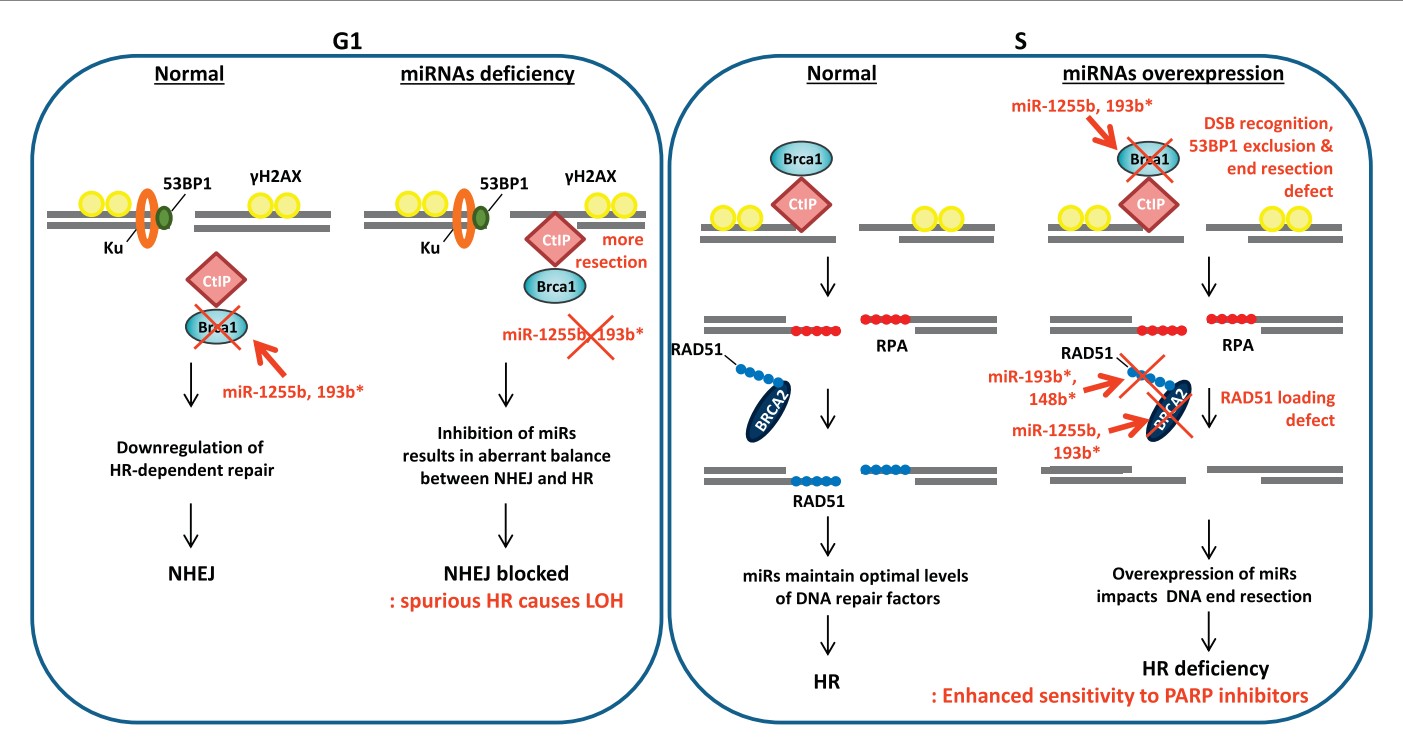

**Figure 8**. A model of miRNA dependent regulation of DSB repair during cell cycle. Model of miRNA-dependent regulation of DSB repair during cell cycle. The balance of HR and NHEJ in dividing cells is crucial for efficient DSB repair. NHEJ is the preferred pathway in the G1 phase with 53BP1 and the Ku complex binding the broken DNA end. miRNAs (such as miR-1255b and miR-193b*) suppress HR factors, particularly BRCA1, preventing end resection of the DNA lesions. However, when these miRNAs are inhibited or deleted it may disrupt the correct choice of DSB repair pathway. Ectopic over expression of BRCA1 will allow CtIP-mediated resection in G1 cells, preventing NHEJ. Furthermore, HR-mediated repair in G1 is detrimental to cell health as it would lead to LOH. In S-phase, DSBs are predominantly repaired by HR and down-regulation of miRNAs targeting BRCA1, BRCA2 and RAD51 may be important in ensuring efficient HR-mediated DSB repair. Over-expression of miRNAs (such as miR-1255b, miR-193b*, and miR-148b*) targeting BRCA1, BRCA2 and RAD51 in the S phase will impede various steps of HR, and the HR deficiency will sensitize these cells to PARP inhibitors.

both the libraries were based on miRbase 14. MDA-MB231 cells were plated on 384-well plate at 1000 cells/well density and reverse transfected with 50 nM control miRNA, miRNAs, or BRCA2 siRNA using Lipofectamine RNAiMax transfection reagent before being treated with 20 µM ABT888 next day. In 3 days, cells were subjected to viability test using Celltiter Glo (#G7572; Promega, Madison, WI). The average luminescence and Z-score was calculated. % viability control was calculated for each miRNA and the cut off was set at 75% control viability and Z-score of positive control, BRCA2 siRNA. Z-score was calculated by the following equation where x is luminescence value for each miRNA, µ is negative control mean, and σ is negative control standard deviation: $z = (x-\mu)/\sigma$.

## Clonogenic assay/colorimetric assay

Clonogenic assays and colorimetric assays with MDAMB231 or 21NT cells treated with PARP inhibitors were done as previously described (*Moskwa et al., 2011*). We modified it to seed 500 cells per six-well plates for clonogenic assay and 1000 cells on 96-well plates for colorimetric assay. Surviving colonies of >1 mm diameter or live cells were quantified.

## Homologous recombination reporter assay

HR assay was performed as described in *Moskwa et al. (2011)*.

## Immunofluorescence

Immunofluorescence in MDAMB231 cells were done as previously described *Lee et al. (2010)* using RAD51 (Santa Cruz, Dallas, TX), H2AX (Cell Signaling, Danvers, MA), γ-H2AX (Cell Signaling), or Cyclin A (Santa Cruz).

## RNA isolation and quantitative real-time PCR

Total RNA was prepared and expression was analyzed by qRT-PCR as described in *Moskwa et al. (2011)*. Gene-specific primers used for qRT-PCR are as follows:

Mre11A-F, TCTGCCTTTAGTGCTGAT GAC, R, GCTCTTCCTCTTTGAGACCC;
RAD50-F, GTGCGGAGTTTTGGAATAGAG, R, GAGCAACCTTGGGATCGTGT;
NBS1-F, CACTCACCTTGTCATGGTATCA, R, CCGTCCTGACAGATCAACATT;
ATM-F, AAGGCTATTCAGTGT GCGAG, R, GGCTCCTTTCGGATGATGGA;
ATR-F, CCAGCATTCTCCAGGTGACA, R, AGCCAGCATTCTCCAACCA T;
ATRX-F, CAAGGTCTGCAAAGAAAGCAG, R, TGGAATCATCATTTTCATCTTCC;
CtIP-F, GGCTTATGTGATCG CTGTGC, R, ACAGCATCAAGCAGCTGAGC;
RPA2-F, TTCACAGGTCACTAT TGTGGG, R, GAACAAAAAGA GCCTGGTAGC;
TOPBP1-F, TTTCCGTGCTGTGG TCTCAC, R, GAAACTCCAGGAC GTCCCAG;
Abraxas-F, TGCA GGAGCATTTTTCAAACC, R, GTATGTCCACTGGTTTTAGCC;
RAD52-F, TA CAATGGCTGGGCACACT C, R, TCCTTCCTTGCCTTCTCCAA;
MDC1-F, GAGACATCTGAGGAAATACAAG, R, TCTTTCTGGTAGCAGTTTCTCA;
BRCA1-F, AGGCAACTTATTGCAGTGTGG, R, ACTTTTCTGGATGCCTCTCAG;
EXO1-F, AGCCAAAGGTGAACCTACTGA, R, AGCTTCCGAGACTTTCCCCT;
RAD51-F, GGGTCGAGGTGAGCTTTCAG, R, GG GCGATGATATTTCCTCCA;
BRCA2-F, GCCAAGTCATGCCACACATT, R, TGTGC CATCTGGAGTGCTT T;
PALB2-F, TTGTTTGTCTCAGCAGGATCTC, R, CTTGGGTGTCATCTGTTCTTT G;
BRIP1-F, ATGAACC AAGGAACTTCACGTC, R, CTGCTGTGTAGTTTCTAAGGGTC.

## Target prediction

We adopted unbiased target prediction algorithms for miRNA-1255b, miRNA-193b*, and miRNA-148b* and each algorithm predicted a list of targets as follows:
Comparative target prediction analysis for miRNA-1255b, miRNA-193b*, and miRNA-148b*:

| | | Number of predicted targets | Number of predicted targets | Number of predicted targets |
|---|---|---|---|---|
| algorithm | URL | miR-1255b | miR-193b* | miR-148b* |
| TargetScan | http://www.targetscan.org/ | 134 | N/A | N/A |
| miRWalk | http://www.umm.uni-heidelberg.de/apps/ zmf/mirwalk/micrornapredictedtarget.html | 906 | 873 | 1352 |
| miRDB | http://mirdb.org/miRDB/ | 131 | 268 | 351 |

The prediction list was compared with an updated list of Human DNA Repair Genes (http://sciencepark. mdanderson.org/labs/wood/dna_repair_genes.html#Human%20DNA%20Repair%20Genes; *Wood et al., 2001*). However, there were no overlapping candidates from the predictions and the DDR genes. Therefore, we used a candidate-based prediction approach using RNA22 (http://cm.jefferson.edu/ rna22v2/) (http://cbcsrv.watson.ibm.com/rna22.html) and PITA (http://genie.weizmann.ac.il/pubs/ mir07/mir07_data.html), to analyze the Human DNA Repair Gene list that resulted in a list of DDR genes predicted as targets of miRNAs of our interest. 18 of them are implicated in HR-mediated repair and they were examined in *Figure 3B*.

## Immunoblots

The immunoblots were done as described previously (*Lee et al., 2010*; *Moskwa et al., 2011*) with BRCA1 (#OP92; Calbiochem, Billerica, MA), BRCA2 (#ab16825-100; Abcam, Cambridge, MA), RAD51 (#sc-8349; Santa Cruz), MDC1 (#sc-27737; Santa Cruz), PalB2 (#A301-246A; Bethyl, Montgomery, TX), BRIP1 (#4578S; Cell Signaling), Exo1 (#sc-19941; Santa Cruz), H2AX (#2595S; Cell Signaling), and γ-H2AX (#9718S; Cell Signaling) antibodies and α-tubulin (#T5168; Sigma, St. Louis, MO) antibodies.

## Immunoprecipitation of miRNA targets

MDAMB231 cells were plated at $0.3 \times 10^6$ cells/well on 6 well-plate overnight and transfected with biotinylated control miRNA (*C. elegans* miRNA [Bi-cel-miR-67]) or biotinylated miR-1255b, miR-193b*,

or miR-148b* (Sigma). The cells were harvested 6 hr after transfection in 700 µl lysis buffer (20 mM Tris-HCl (pH 7.5), 100 mM KCl, 5 mM MgCl$_2$, 0.3% IGEPAL CA-630) containing freshly added 300U RNaseOUT (Invitrogen, Grand Island, NY) and Protease Inhibitor Cocktail (Roche, South San Francisco, CA) and incubated on ice for 20 min. After centrifugation for 15 min at 10000×$g$, 4°C, a 50-ml aliquot of supernatant was taken as input for subsequent RNA extraction. The remaining supernatant was incubated with activated Streptavidin-Dyna beads (Dynabeads M-280 Streptavidin, #112.05D; Invitrogen) for overnight at 4°C. Reaction mixture was centrifugated to remove unbound material. The beads were washed five times with lysis buffer and treated with 10U DNase I in lysis buffer for 10 min at 37°C. The beads were washed and treated with 50 µg/100 µl Protease K in 10% SDS containing lysis buffer for 20 min at 55°C. After centrifugation, the supernatant was taken for RNA extraction using acid phenol-choloroform (Applied Biosytems). RNA was subjected to qRT-PCR using gene specific primers. The analysis was done as follows: miRNA pull-down/control pull-down ('A'), miRNA input/control input ('B'); fold enrichment = A/B.

## Luciferase assay

The wild type (WT) or mutant (Mt) miRNA recognition elements (MREs) of target genes were synthesized as oligo sequences, annealed and cloned in psiCHECK2 (Promega) downstream to *Renilla* luciferase. Luciferase assay in MDAMB231 cells using WT and Mt MRE constructs was done as described previously (*Moskwa et al., 2011*). The oligonucleotide sequences are as follows:

**BRCA1** MRE1+2+3-F, TCGAAAGATGAAGTTTCTATCATCCAGAAGACCTACA TCAGGCCTTCATC CTGAAGCCAGGAGTGGAAAGGTCATCCC, R, GGCCGGGA TGACCTTTCCACTCCTGGCTTCAG GATGAAGGCCTG ATGTAGGTCTTC TGGA TGATAGAAACTTCATCTT; **BRCA2** MRE1-F,TCGACA ATCAACAAAATGGTCATCCA, R, GGCCTGGATGA CCATTTTGTTGATTG; **BRCA1** MRE4+5+6-F, TCGAAGTGTGCAGCATTTGAAAACCCGAAGGCA AGATCTAGAGGGAACCCCTGAAGCTGGCC AACATGGTGAAACCCC, R, GGCCGGGGTTTCACCATGTTGGCCAGC TTCAGGGGTTCCCTCTA GATCTTGCC TTCGGGTTTTCAAATGCTGCACACT; **BRCA2** MRE2-F, GGCCAGGGGTTA TATGAC TATTTTTA, R, TCGATTGGCCAAG GTGGTGAAATCCC;
**RAD51** MRE1-F, TCGATTGGCCAAGGTGGTGA AATCCC, R, GGCCGGGATTTC ACCACCTTG GCCAA;
**RAD51** MRE2+3-F, TCGATCTTATGTTTCCAAGAGAACTAG AAGTCTCTACAAA AAATGCAGAACT, R, GGCCAGTTCTGCATTTTTTGTAGAGACTTCTAGTTCTCT TGGAAACATA AGA.

The oligonucleotides for mutant MREs are as follows:

**Mt BRCA1** MRE1+2+3-F, TCGATTGATGAAGAA ACGAGCAGGCAGAAGTGGTA CAAGAGGC GAGCAGGCTGAAGCCTGGTGAGGAAACGAGTAGGC, R,GGCCG CCTACTCGTTTCCTCACC AGGCTTCAGCCTGCTCGCCTCTTGTACCACTTCTGCCTGCTCGTTTCTTCATCAA;
**Mt BRCA2** MRE1-F, TCGACAATGAACAAAATGGGCAGGCA,R, GGCCTGCCTGCCCATTTTGTT CATTG;
**Mt BRCA1** MRE4+5+6-F, TCGAAGTGAGCAGGAGTTTTAAAGCCGAAGGGTAG AGCGAGAGGG AACCCCTGAAGCAGTCGA ACAGGGTTTTAGGGC, R,GGCCGC; CCTAAAACCCTGTTCGACTGCTTCAGGGGTTCCCTCTCGCTCTACCCTTCGGC TTTAAAACTC CTGCTCACT;
**Mt BRCA2** MRE2-F, TCGAGTAAAATAGGCATTTTAGGGCT, R, GGCCAGCCCT AAA ATGCCTATT TTAC;
**Mt RAD51** MRE1-F, TCGATAGGGCAAGGAGTTTTAAGGCC, R, GGCCGGCCTT AAAACTCCT TGCCCTA;
**Mt RAD51** MRE2+3-F, TCGATCATATGTTGCCTTGAGAACTAGAAGACTCTACA TATTATCCTGAACT, R, GGCCAGTT CAGGATAATATGTAGAGTCTTCTAGTTC TCAAGGCAACATATGA.

## Cell cycle synchronization

MDAMB231 or MCF10A cells were seeded at 0.2 × 10$^6$ cells/well and treated with 500 µM mimosine for 24 hr. The cells were washed and released into growth media and collected for FACS analysis and RNA extraction after indicated time intervals. For FACS, the cell were fixed in 70% ethanol, washed with PBS buffer, and analyzed in PI/RNase staining buffer (BD PharMingen, San Jose, CA). MDAMB231 cells transfected with miRNA antagomirs were similarly synchronized 48 hr after transfection. The siRNA for CtIP was obtained from Dharmacon, Pittsburgh, PA (#J-011376-06).

## Statistical analysis of genomic data

### TCGA

We obtained processed LOH and copy number calls for serous ovarian cancer samples from TCGA (2011) (*Cancer Genome Atlas Research Network, 2011*). In brief, as parts of the TCGA initiative, the LOH analysis was performed using Human HapMap 1M Duo beadchips at the Hudson Alpha Institute for Biotechnology, and the copy number analysis was performed using Agilent Human Genome CGH 244A microarray at Memorial Sloan Kettering Cancer Center. We marked those LOH events as heterozygous deletion mediated, which overlapped with deletion events (aCGH log2 signal intensity ratio $<-0.20$) for at least 80% of its length. After excluding those LOH events, we focused on the copy neutral LOH events of size >1 Mb. There were 26,176 LOH events in 418 samples. In each sample, at each of the three miRNA loci, we inferred its copy number status as follows: miRNA loci of log2 signal intensity $<-0.2$ were flagged as deletions, and those regions with signal intensity $\geq-0.2$ to $\leq 0.2$ were flagged as copy neutral. We also obtained processed expression data for the same samples analyzed using the Affy U133A array at the Broad Institute from the TCGA data portal (PMID: 21720365).

### DF/HCC cohort

The processed LOH calls and per SNP copy number calls were previously described (GSE39130) (*Wang et al., 2012b*). We focused only on the LOH events with size >10 kb; we counted the number of marker SNPs in those LOH region (N), and the number of such SNPs with allelic copy number <1.9 (n). If n/N was $\geq 0.8$, we postulated that LOH to be deletion mediated; while the remaining ones (n/N <0.8) as copy neutral LOH. There were 685 copy neutral LOH events in these samples. We repeated the analyses with a n/N cut-off of 0.75. In each sample, at each of the three miRNA loci, we inferred its copy number status based on the nearest marker SNP. If the copy number call was <1.9, we marked that as a deletion, and if the copy number call was between 1.9 and 2.1, we flagged it as copy neutral, respectively.

## Acknowledgements

The RPE1-Fucci cells were a gift from David Pellman's laboratory (DFCI). We thank David Livingston's group (DFCI) for sharing unpublished results on miR-545 with us. The ABT888 used in all our studies was provided by Abbott Laboratories.

## Additional information

### Funding

| Funder | Grant reference number | Author |
|---|---|---|
| National Institutes of Health | R01CA142698 | Dipanjan Chowdhury |
| American Cancer Society | RSG-12-079-01 | Dipanjan Chowdhury |
| National Institutes of Health | R01HL52725 | Alan D'Andrea |
| Susan Komen Foundation | KG101186 | Eunmi Park |
| Ann-Fuller Foundation | | Dipanjan Chowdhury |
| Susan Smith Women's Cancer Grant | | Dipanjan Chowdhury |
| Mary Kay Foundation | | Dipanjan Chowdhury |

The funders had no role in study design, data collection and interpretation, or the decision to submit the work for publication.

### Author contributions

YEC, Acquisition of data, Analysis and interpretation of data, Conception and design, Drafting or revising the article; YP, Conception and design, Acquisition of data; EP, Conception and design, Acquisition of data, Analysis and interpretation of data; PK, SD, Acquisition of data, Analysis and interpretation of data; AD, Analysis and interpretation of data, Drafting or revising the article; DC, Conception and design, Analysis and interpretation of data, Drafting or revising the article

## Ethics

Human subjects: The clinical data from patients was obtained via published sources which include the Cancer Genome Atlas.

## Additional files

### Major dataset

The following previously published dataset was used:

| Author(s) | Year | Dataset title | Dataset ID and/or URL | Database, license, and accessibility information |
|---|---|---|---|---|
| Wang ZC, Birkbak NJ, Culhane AC, Drapkin R, Liu J, Fatima A, Tian R, Daniels KE, Piao H, Miron A, Quackenbush J, Berkowitz RS, Iglehart JD, Matulonis UA | 2012 | DF/HCC Cohort | GSE39130; http://www.ncbi.nlm.nih.gov/geo/query/acc.cgi?acc=GSE39130 | Publicly available at GEO (http://www.ncbi.nlm.nih.gov/geo/). |

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
