## [Decision Letter]

Thank you for sending your work entitled “MicroRNAs down-regulate homologous recombination in the G1 phase of cycling cells to maintain genomic stability” for consideration at *eLife*. Your article has been favorably evaluated by a Senior editor and 3 reviewers, one of whom, Timothy Nilsen, is a member of our Board of Reviewing Editors.

The Reviewing editor and the other reviewers discussed their comments before we reached this decision, and the Reviewing editor has assembled the following comments to help you prepare a revised submission.

There was some difference of opinion regarding the perceived impact of the work but in the end we have agreed to consider a revised manuscript that addresses the following points.

Two referees thought that the manuscript was not well written and as a result was confusing. First, the writing does not meet the standard of scientific English for publication. This is reflected by frequent grammatical errors and informal phrases throughout the text. Second, the writing does not flow smoothly, as many concepts or findings are not explained well. Some data, for instance, the rescue experiments in Figure 5, can only be judged after getting a better understanding of what the authors want to deliver. The weakness in the writing significantly compromised the quality of this work. We suggest rewriting this paper in a precise and concise way, and explain the rationale clearly before moving on to the experimental details. We further suggest that you consult a senior colleague to offer comments on your revised text.

Additional specific comments:

1) Criteria for selection of positives in the screen

In the first paragraph of the Results section, the authors claim “we screened two independently synthesized commercially available miRNA mimic libraries and focused on overlapping 'hits'.” Having said that, it is not entirely clear what criteria was used to select those miRNAs that scored positively in this screen.

The authors selected miR1255b, miR-148b* and miR-193b* for further studies. Maybe, I am missing something here, but why were miR-185*, miR-367*, and miR-1204 *not* selected?

Also, why were miRNAs that caused a loss of viability equal to BRCA2 siRNA were not considered?

2) In Figure 3, it would be important to include a few miRNAs that scored negatively in their screen.

3) In Figure 4, it would be important to show the luciferase assays with the wild-type 3'UTR in the presence of an inhibitor of the miRNA under study (e.g., miR-1255b on the upper panel).

4) The problem of HR silencing during G1 is not solved here but pushed back one step: i.e., there is no information as to how the levels of miRNAs are regulated in a cell cycle dependent manner.

5) The manuscript itself meanders around such that the reader can lose interest.

6) Much of the data is borderline in quality: i.e., affinity pull downs, luciferase assays, etc.

7) In mRNAs apparently regulated by more than one miRNA, why wasn't the combined effect measured?

8) Many of the experiments would have been much more convincing if done in synchronized or FACS-sorted cells.

---

## [Author Response]

We thank the reviewers for their constructive and insightful suggestions and comments. Addressing these comments has significantly improved the manuscript.

Specific comments:

*1) Criteria for selection of positives*
*in the screen*

*In the,first paragraph of the Results section, the authors claim “we screened two independently synthesized commercially available miRNA mimic libraries and focused on overlapping 'hits'.” Having said that, it is not entirely clear what criteria was used to select those miRNAs that scored positive in this screen*.

First of all, I would like to apologize for the confusion regarding the screen and our criterion for selecting specific miRNAs. We have completely re-written this section to clarify these issues. To systematically identify miRNAs that impact PARP inhibitor sensitivity we screened two commercially available miRNA mimic libraries (Qiagen and Applied Biosytems). MiRNA mimics are 20-22 nts, chemically modified double-stranded RNA molecules designed to mimic endogenous mature miRNAs. The miRNA mimics from Qiagen and Applied Biosystems have proprietary chemical modifications that cause inactivation of the ‘passenger’ strand, allowing the ‘active’ strand to associate with target transcripts and regulate gene expression. We used the Applied Biosytems library as our primary screen, and this yielded a rank ordered list of the top 13 miRNAs based on viability percentage that sensitized cells to PARP inhibitor. Prior to individual validation experiments we utilized the results from the screen performed with the Qiagen library to confirm the impact of these miRNAs on ABT888 sensitivity. Basically the screen with the Qiagen library was used to validate the ‘hits’ from the Applied Biosystems library, before we did any focused experiments.

*Also,*
*why were miRNAs that caused a loss of viability equal to BRCA2 siRNA were not considered?*

BRCA2 siRNA served as a positive control for each plate as the original discovery of PARP inhibitors was made using BRCA2 mutant cells. Depletion of BRCA2 dramatically sensitizes cells to PARP inhibitors. However, BRCA2 is an important gene for repair of replication-induced DNA damage and depletion of BRCA2 has an impact on viability even in the absence of PARP inhibitors (Figure 1—figure supplement 1). Our goal was to identify miRNAs that cause loss of viability only when cells are treated with PARP inhibitors. Therefore, miRNA mimics that caused significant loss of viability in the absence of PARP inhibitors were not considered. We used BRCA2 depletion in untreated cells as a threshold to eliminate miRNAs that may cause loss of viability independent of PARP inhibitors.

*The authors selected miR1255b, miR-148b* and miR-193b* for further studies. Maybe, I am missing something here, but why were miR-185**, *miR-367* and miR-1204* not *selected?*

The results in Figures 1 and 2 were derived by artificially introducing miRNA mimics in cells. To explore the physiological relevance of the seven miRNAs that significantly impacted HR-mediated DSB repair and PARP inhibitors sensitivity we needed to identify cells where these miRNAs are endogenously expressed. From the perspective of cancer biology, the HR pathway is extremely relevant for breast tumors, specifically triple negative breast cancer (TNBC) and clinical trials with PARP inhibitors are underway in TNBC. Therefore in Figure 3 we initially assessed the endogenous expression of seven selected miRNAs (miR-1231, miR-1255b, miR-148b*, miR-876-3p, miR-221*, miR-193b*, and miR-185*) in a panel of TNBC lines. Of the seven miRNAs, only miR-1255b, miR-148b*, and miR-193b* had detectable and aberrant over expression in TNBC lines (Figure 3). Therefore we focused on an in-depth analysis of miR-1255b, miR-148b*, and miR-193b*. Again this section has been re-written and the point clarified.

*2) In*
Figure 3*, it would be important to include a few miRNAs that scored negatively in their screen*.

The rationale for Figure3A was not clearly explained in the original version of the manuscript, and we suspect that this comment may be due to this lack of clarity on our part. ∼880 miRNAs scored ‘negatively’ in our screen. The expression level of a large majority of these miRNAs is not known in TNBC lines. I’m not sure what criterion we would utilize to select the ‘negative’ miRNAs and look at their expression in TNBC lines, and more importantly how that would add to our story. However, we did take the reviewers’ comment in mind to further substantiate this figure. Of the seven miRNAs we initially selected based on impact on HR-mediated DSB repair, miR-1231, miR-876-3p, miR-221*, and miR-185* were not expressed in the TNBC lines. However, it is important to show that our detection method is valid for these miRNAs, and their lack of expression in TNBC lines is not an artifact. Therefore we have included data showing that these miRNAs are expressed in other cell lines.

*3) In*
Figure 4*, it would be important to show the luciferase assays with the wild-type 3'UTR in the presence of an inhibitor of the miRNA under study (e.g. miR-1255b on the upper panel)*.

We thank the reviewers for this suggestion, and have conducted the luciferase assays in the presence of the inhibitors (Figure 4). Consistent with the other results, inhibition of miR-1255b enhanced luciferase activity of the BRCA1 and BRCA2 reporter construct, inhibition of miR-148b* enhanced luciferase activity of the RAD51 reporter construct, and inhibition of miR-193b* enhanced luciferase activity of the BRCA1, BRCA2, and RAD51 reporter constructs.

*4) The problem of HR silencing during G1 is not solved here but pushed back one step: i.e., there is no information as to how the levels of miRNAs are regulated in a cell cycle dependent manner*.

We completely agree with the reviewers and we have mentioned this point in the Discussion. Looking at how the miRNAs are regulated in a cell cycle dependent is beyond the scope of this study, and would definitely be an important follow-up of this work.

*5) The manuscript itself meanders around such that the reader can lose interest*.

One issue here is that we have covered significant ground in this one manuscript, and gone from an unbiased screen in cell lines, to molecular mechanism of multiple ‘hits’ and then to clinical analysis. That being said, we have revised the text to improve how the manuscript reads.

*6) Much of the data is borderline in quality: i.e., affinity pull downs, luciferase assays, etc*.

For the affinity pull down assays we have added additional data using combination of biotinylated miRNA mimics with overlapping targets to confirm and improve the results presented. For the luciferase assays based on the reviewers comments we have used the loss-of-function approach to further substantiate our previous results.

*7) In mRNAs apparently regulated by more than one miRNA,*
*why wasn't the combined effect measured?*

We thank the reviewers for this suggestion. We have now done a combination of these miRNAs with overlapping targets and the results are very consistent. First for the affinity pulldown with biotinylated miRNA mimics we used a combination of miR-1255b and miR-193b*, and a combination of miR-148b* and miR-193b* in Figure 3. As anticipated relative to the individual biotinylated mimics there is increased amount of BRCA1 and BRCA2 transcripts immunoprecipitated with the miR-1255b/miR-193b*combination and increased RAD51 mRNA immunoprecipitated with the miR-148b*/miR-193b* combination. Second, we used combined inhibition of miRNAs with overlapping targets to observe a synergistic increase in target transcripts in G1 (Figure 5). There is a significant increase in BRCA1 and BRCA2 transcripts in cells co-transfected with miR-1255b ANT and miR-193b*ANT, and a significant increase in RAD51 mRNA in cells co-transfected with miR-148b*ANT and miR-193b*ANT.

*8) Many of the experiments would have been much more convincing if done in synchronized or facs sorted cells*.

In Figure 5 we first discovered that these miRNAs are regulated during the cell cycle. All the experiments shown in Figure 5 were conducted in synchronized cells. We were concerned that synchronizing cells can influence the DNA damage response, and therefore in Figure 6 we tried to observe cell cycle phase specific effects without synchronizing cells. Staining cells with Cyclin A allowed us to distinguish the S/G2 versus the G1 cells, and this is a standard procedure in the DNA damage response field. In Figure 6 we utilized his system, and then we used the Fucci system (which has been better explained in revised manuscript) to yet again visually distinguish G1 and S/G2 cells for the functional experiments (Figure 6).